# PCSK9 and Other Metabolic Targets to Counteract Ischemia/Reperfusion Injury in Acute Myocardial Infarction and Visceral Vascular Surgery

**DOI:** 10.3390/jcm11133638

**Published:** 2022-06-23

**Authors:** Silvia Ortona, Chiara Barisione, Pier Francesco Ferrari, Domenico Palombo, Giovanni Pratesi

**Affiliations:** 1Vascular and Endovascular Surgery Unit, IRCCS Ospedale Policlinico San Martino, Largo Rosanna Benzi, 10, 16132 Genoa, Italy; silviaortona@gmail.com (S.O.); domenico.palombo@unige.it (D.P.); giovanni.pratesi@unige.it (G.P.); 2Department of Surgical and Integrated Diagnostic Sciences, University of Genoa, Viale Benedetto XV, 6, 16132 Genoa, Italy; 3Department of Civil, Chemical and Environmental Engineering, University of Genoa, Via Opera Pia, 15, 16145 Genoa, Italy; pier.francesco.ferrari@unige.it; 4Research Center for Biologically Inspired Engineering in Vascular Medicine and Longevity, University of Genoa, Via Montallegro, 1, 16145 Genoa, Italy

**Keywords:** ischemia/reperfusion injury, acute myocardial infarction (AMI), ischemic stroke, vascular surgery, acute kidney injury (AKI), PCSK9, myostatin (MSTN), adipokines, PPARs

## Abstract

Ischemia/reperfusion (I/R) injury complicates both unpredictable events (myocardial infarction and stroke) as well as surgically-induced ones when transient clampage of major vessels is needed. Although the main cause of damage is attributed to mitochondrial dysfunction and oxidative stress, the use of antioxidant compounds for protection gave poor results when challenged in clinics. More recently, there is an assumption that, in humans, profound metabolic changes may prevail in driving I/R injury. In the present work, we narrowed the field of search to I/R injury in the heart/brain/kidney axis in acute myocardial infarction, major vascular surgery, and to the current practice of protection in both settings; then, to help the definition of novel strategies to be translated clinically, the most promising metabolic targets with their modulatory compounds—when available—and new preclinical strategies against I/R injury are described. The consideration arisen from the broad range of studies we have reviewed will help to define novel therapeutic approaches to ensure mitochondrial protection, when I/R events are predictable, and to cope with I/R injury, when it occurs unexpectedly.

## 1. Introduction

Ischemia/reperfusion (I/R) injury is a multifactorial process that occurs when the blood supply to organs and tissues is temporarily interrupted. Many studies confirm that the restoration of blood flow increases tissue damage more than ischemia itself; therefore, different efforts have been made in order to minimize reperfusion damage. I/R injury constitutes an ominous clinical problem in different organs, such as the heart, causing acute myocardial infarction; the brain, causing stroke; the kidneys, causing renal damage; and the intestines, causing intestinal infarction and multi-organ failure. This two-step phenomenon can also be induced by surgical processes, such as during cardiac surgery, cardiopulmonary bypass, and major vascular or visceral organ interventions, thereby leading to acute renal failure during organ transplantation, which contributes to acute graft failure and rejection, and, in Tourniquet-related surgery, causing deep vein thrombosis and pulmonary embolism. Therefore, peri-surgical time is characterized by the activation of stress-response signals involving inflammatory, endocrine, metabolic, and immunological mediators (Figure 1) [1].

## 2. Method

This review analysis studies Proprotein Convertase Subtilisin/Kexin type 9 and other metabolic targets such as Myostatin, Adipokines, and Peroxisome Proliferator Activated Receptors, with a focus on their role in Ischemia/reperfusion injury.

Searches in PubMed, Google Scholar, and university textbooks (Pontieri, G.M.; Russo, M.A.; Frati, L. *Patologia generale*) were carried out with several different combinations of keywords:−(“Ischemia Reperfusion Injury” [Review] OR “Ischemia Reperfusion Damage” [Review] OR “surgical Ischemia Reperfusion Injury” [Review] OR “Myocardial Ischemia Reperfusion” [Review] OR “Kidney Ischemia Reperfusion” [Review] OR “Heart Ischemia Reperfusion” [Review] OR “Ischemic Stroke [Review]” OR “Vascular surgery” [Review] OR “Acute Kidney Injury” [Review]) first evaluating the most recent reviews which better summarise the previous ones.−((“Proprotein Convertase Subtilisin/Kexin type 9” [All Fields] OR “PCSK9” [All Fields]) AND (“Ischemia Reperfusion Injury” [All Fields]))−((“Proprotein Convertase Subtilisin/Kexin type 9 inhibitors” [All Fields] OR “PCSK9 inhibitors” [All Fields]) AND (“Ischemia Reperfusion Injury” [All Fields]))−((“Myostatin” [All Fields] OR “Mstn” [All Fields]) AND (“Ischemia Reperfusion Injury” [All Fields]))−(“Myostatin inhibitors” [All Fields] OR “Mstn inhibitors” [All Fields])−((“Adipokines” [All Fields]) AND (“Ischemia Reperfusion Injury” [All Fields]))−((“Adipokines inhibitors” [All Fields]) AND (“Ischemia Reperfusion Injury” [All Fields]))−((“Peroxisome Proliferator-Activated Receptors” [All Fields] OR “PPARs” [All Fields]) AND (“Ischemia Reperfusion Injury” [All Fields]))−((“Peroxisome Proliferator-Activated Receptors Agonists” [All Fields] OR “PPARs Agonists” [All Fields]) AND (“Ischemia Reperfusion Injury” [All Fields]))−((“Ischemia Reperfusion” [All Fields]) AND (“miRNA” [All Fields]))−((“Ischemia Reperfusion” [All Fields]) AND (“Nanoparticles” [All Fields])) −((“Ischemia Reperfusion” [All Fields]) AND (“Biomaterials” [All Fields]))−((“Ischemia Reperfusion” [All Fields]) AND (“Nanoparticles” [All Fields]) AND (“Pioglitazone” [All Fields]))

## 3. Cellular Pathology in Ischemia and Reperfusion

In the cell, ischemia causes oxygen deprivation and blockage of mitochondrial oxidative phosphorylation, which decreases ATP production and increases the concentration of lactic acid; these effects result in an altered function of the ATP-dependent ion pump, favoring the entry of calcium, sodium, and water into the cell that ultimately results in cellular acidosis, edema, and swelling [2,3]. This condition has deleterious consequences on organ function at different extents depending on the cell types; for instance, in endothelial cells lining microscopic blood vessels, it may lead to impaired barrier function increasing vascular permeability and leakage [4]. All this happens at the same time as enzymatic alterations that profoundly affect the composition of cell membranes. The activation of calcium-dependent phospholipase causes the dissociation of lipoproteins and the release of triglycerides and fatty acids [5].

Following ischemia, reperfusion allows the essential substrates for the generation of ATP to replenish (such as glucose or free fatty acids), to increase the available oxygen, and to normalize the extracellular pH. These are crucial factors for the survival of the tissue, but can also contribute to the exacerbation of damage in the area [4]. 

Specifically, with the restoration of blood flow, the pH normalizes. The following H^+^ gradient across the plasma membrane causes the intracellular accumulation of Ca^2+^ [6], which promotes events, including the activation of lipases, nucleases, and proteases, that undermine the cellular structure (e.g., reactivation of calpain, a calcium-dependent protease whose action is also inhibited by the drop in pH).

The sudden restoration of aerobic metabolism is accompanied by the accumulation of reactive oxygen species (ROS)—in particular, superoxide anion (O_2_^•−^), and reactive nitrogen species (RNS). Under physiological conditions, (O_2_^•−^) is converted to hydrogen peroxide (H_2_O_2_) by superoxide dismutase (SOD) and is subsequently inactivated by catalase into H_2_O and O_2_. During I/R, xanthine dehydrogenase^−^derived xanthine oxidase utilizes hypoxanthine or xanthine as a substrate and O_2_ as a cofactor to produce (O_2_^•−^) and uric acid, with a consistent production of ROS that generates an excessive amount of highly unstable hydroxyl radicals (HO•) [7] (Figure 2).

I/R injury is associated with the impaired function of mitochondria, as they supply energy and regulate the ion balance and thermogenesis in the cell [8]. 

Mitochondrial dysfunction causes cell/organ injury through several mechanisms, including diminished cellular energy status (low cellular ATP level, energy stress), enhanced production of ROS—including (O_2_^•−^), H_2_O_2_, HO•—and peroxynitrites [9] with the subsequent development of oxidative stress. Furthermore, mitochondrial damage triggers the intrinsic apoptotic pathway by release of mediators leading to caspase-3 activation and programmed cell death, thereby fueling the feed-forward loop of tissue damage and organ failure [8,10,11,12].

**Figure 2 jcm-11-03638-f002:**
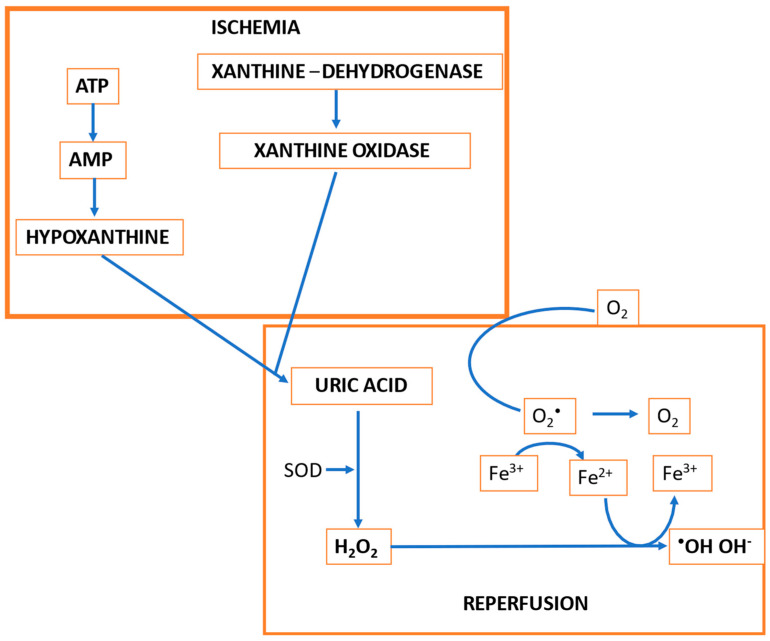
Biochemical events triggering ischemia/reperfusion injury (adapted with permission from Ref. [13] 2022, *Piccin Editore*).

## 4. Ischemia/Reperfusion in Acute Myocardial Infarction and Aftermaths in Brain and Kidneys

I/R injury is a cellular event that may potentially affect all the organs. This section is focused on acute myocardial infarction (AMI), an ischemic phenomenon evoked in more than half of the cases by coronary artery disease (CAD), leading to myocardial necrosis [14,15]; the side effects of AMI on the brain and kidneys, as downstream target organs, will be revised as it may represent a primum movens of ischemic stroke or cause downstream effects of renal failure, two comorbidities often constituting the cause of long-term complications and mortality after AMI.

European epidemiology updated for 2016 indicates that cardiovascular diseases (CVDs) remain the most common cause of death in Europe, including new statistics for mortality, morbidity, and treatment. CVDs cause more than 4 million deaths each year across Europe, accounting for 45% of all deaths; of them, 20% are attributable to coronary heart disease, including AMI; 11% to cerebrovascular disease, as in the case of stroke; and 14% to other CVDs [16].

The pathobiology of AMI is mainly based on the rupture or erosion of a vulnerable, lipid-laden atherosclerotic coronary plaque, resulting in exposure to the circulating blood of a highly thrombogenic plaque core and matrix materials that cause platelet activation and thrombus generation. Heart ischemia from reduced coronary blood flow leads to myocardial cell injury or death–myocardial necrosis begins after as little as 20 min of coronary occlusion, ventricular dysfunction, and cardiac arrhythmias [14].

In a wide spectrum of diseases based on ischemia and reperfusion, such as heart and brain I/R injury, heart failure, and inherited diseases, mitochondria play a crucial role as they regulate cell homeostasis and provide viability through ATP production, ion balance and signaling pathways, homeostasis, thermogenesis, and programmed cell death [8].

The protection of the mitochondrial pool and its function is a promising therapeutic approach to prevent cell/organ dysfunction in response to pathological stimuli, especially when considering organs with high energy demands such as the heart—where mitochondria occupy about 35% of the volume of adult cardiomyocytes and provide about 90% of the ATP through oxidative phosphorylation [8].

The IMMEDIATE trial, which tested the out-of-hospital administration of intravenous Glucose–Insulin–Potassium (G-I-K) in patients with suspected acute coronary syndromes, proved the safety and the potential benefit of G-I-K administration in the setting of heart ischemia by providing sarcolemmal and mitochondrial protection. Compared with placebo, G-I-K administration was not associated with an improvement in 30-day survival, however, it was associated with lower rates of the composite outcome of cardiac arrest (CA) or in-hospital mortality [17].

To date, together with Percutaneous Coronary Intervention (PCI), several new therapeutic strategies, such as reducing inflammation, mitigating reperfusion injury, inducing myocardial regeneration, and ameliorating adverse remodeling, are under active investigation, but, with the exception of ACE inhibition, none of them were proved of benefit in the acute care setting [14]. Moreover, when considering the long-term outcome, AMI-evoked comorbidities, such as ischemic stroke and renal failure, must be taken into consideration (Figure 3).

AMI/STROKE: Stroke is defined as rapidly developing clinical signs of focal (or global) disturbance of cerebral function lasting for 24 h (unless interrupted by surgery or death) with no apparent cause other than a vascular origin (i.e., hemorrhage); it is a rare but feared complication of AMI. In most cases, a stroke after an AMI is of ischemic origin, although during the first 24 h it might be a hemorrhagic consequence of thrombolytic therapy [18]. The pathophysiology of stroke after myocardial infarction is unclear. One hypothesis recognizes a common mechanism in embolism from a left ventricular thrombus. However, it can explain only a small fraction of myocardial infarction–related strokes. Other important processes may occur, such as in situ thrombosis and artery-to-artery embolism because of atherosclerosis and unfavorable hemodynamics. Moreover, an AMI is followed by increased fibrinogen levels and a pronounced sympathetic activation, which may facilitate thrombus formation in patients with atherosclerosis in the aorta and the cervical and cerebral arteries [18].

Ischemic stroke burdens 0.9% of patients within 1 month and 3.7% within a year after an AMI, with a doubled 1-year mortality compared with those not complicated with stroke. A retrospective large cohort study utilized a representative 5% sample of Medicare claims data from 2008 to 2015, focusing on the endpoint of stroke caused by ischemia. The risk of ischemic stroke was highest—almost 3-fold—during the first 4 weeks after AMI, but remained enhanced up to week 12, indicating that the risk of ischemic stroke appears to extend beyond the considered window of a 1-month risk period from AMI [19].

The combined use of oral anticoagulants and dual antiplatelet therapy with aspirin in addition to PCI is associated with a significant reduction of cardiovascular death, AMI, or stroke—as compared with a standard dual antiplatelet therapy regimen—but with a drawback of a 2-fold increase in the risk of major bleeding [20].

AMI/RENAL FAILURE: The incidence of acute kidney injury (AKI) due to coronary revascularization has been increasing worldwide [21,22]. Clinical and experimental studies have demonstrated that myocardial I/R causes AKI [22,23] through a multifaceted mechanism comprised of hypo- or loss of pulsatile perfusion, hemolysis, and a systemic inflammatory response. These factors may increase the ROS production that is the leading cause of I/R injury in kidneys, although the underlying mechanism of kidney damage following AMI remains to be fully elucidated [24]. In a swine model, AMI induced subclinical AKI early after hemodynamic instability, with a transient increase of the tubular damage biomarkers that declined after interventions; such fluctuation of signs from tubular damage without glomerular loss of function may indicate appropriate timing for effective renoprotections [25].

The heart/kidney axis may also have an opposite direction, as renal failure is a recognized dependent risk factor for coronary events and myocardial I/R injury. The incidence of AMI among chronic kidney disease (CKD) patients is more than twice that of individuals without CKD. CKD patients have a high prevalence of atheromatous and/or non-atheromatous CVD; moreover, oxidative stress, systemic inflammation, and the accumulation of uremic toxins secondary to renal dysfunction, which are endogenous compounds with vasoactive properties normally cleared by the kidney, [26] are some of the factors that contribute to heart damage. 

Despite various strategies for cardioprotection, a lot remains to be explored for the prevention of CKD-induced myocardial susceptibility to I/R injury. A preclinical study on mice demonstrated that renal failure induced by 5/6 nephrectomy significantly aggravated the cardiac injury after I/R with respect to non-nephrectomised mice, leading to more severe cardiac dysfunction and increased myocardial infarct size, with a higher expression of endoplasmic reticulum (ER) stress-mediated apoptotic proteins. Treatment with the chemical ER chaperone sodium 4-phenylbutyrate, commonly used in protein misfolding diseases, ameliorated cardiac dysfunction and lessened the infarct size and myocardial apoptosis after I/R injury in mice with CKD. This evidence indicates the involvement of excessive activation of ER stress-mediated apoptosis in CKD-induced myocardial susceptibility to I/R injury, suggesting new pharmacological tools that can counteract myocardial I/R injury in the setting of CKD [27].

## 5. Current Practice to Counteract Ischemia/Reperfusion Injury in Vascular Surgery 

Heart and renal impairment on an I/R basis may not only be due to unexpected events, as in the case of myocardial infarction, but are often the consequences of predictable procedures, such as the aortic cross clampage needed for open surgical repair of abdominal aortic aneurysm (AAA), where they constitute the major causes of perioperative morbidity and mortality. Indeed, although multiple risk factors for myocardial and renal failure are often present in patients undergoing AAA open surgery, thereby differently affecting their susceptibility, a common trigger of damage is represented by hemodynamic changes occurring at the interruption and at the restoration site of the blood flow (Figure 4). The need for a blood supply is organ-specific, therefore different protective strategies are undertaken in combination during the surgery.

In the SICVE guidelines, the clinical complications due to transient surgical clamping of the supra renal aorta and the recommendations to reduce the damage to specific organs are reported [28].

Cardiac events are the leading cause of peri-operatory and long-term complications after elective AAA repair, with a rate between 2.2% and 15%; although pre-operative significant CAD has been revealed in up to 70% of patients undergoing AAA surgery, large, randomized clinical trials show no evidence in favor of pre-operative coronary revascularization. Recent reports highlight a rise in cardiac troponin I in 30% of elective AAA repairs and suggest the use of β-blockers to reduce cardiac complications, as recommended by the European Society of Cardiology and by the American Heart Association [29].

Renal injury in the setting of vascular surgery is another cause of morbidity and mortality after elective AAA repair. Although the multifactorial etiology of AAA comprises the use of nephrotoxic agents (antibiotics, anesthesia, contrast media, diuretics, and myoglobin) and a pre-existing renal impairment, I/R injury remains one of the leading causes of acute renal failure post-aortic intervention; indeed, the duration and position (i.e., superior to the mesenteric artery) of the aortic clamping represent important predictors for organ damage. The risk for post-operative renal failure has been shown to be minimized when suprarenal aortic cross-clamping lasts less than 20 min, while it increases by 10-fold if the clamping time is higher than 50 min. The optimal site for aortic clampage is still debated and is dependent on the position and morphology of the AAA [30].

Hypothermia is the most widely recognized intervention in clinical practice to restrain renal complications by reducing cell metabolism, metabolic consumption of oxygen, and ATP. Intra-operative renal cooling is obtained by the perfusion of kidneys with cold media, such as Ringer’s solution and Histidine-Tryptophan-Ketoglutarate (HTK) solution (Custodiol^®^), a cardioplegic medium originally employed to induce cardiac stasis and to protect the myocardium during open-heart surgery [31].

Pharmacological options to improve renal blood flow have been investigated: mannitol, which is freely filtered and does not undergo tubular reabsorption, is used as an osmotic diuretic to reduce renal vascular resistance and thereby increase renal blood flow—in combination or as an alternative to diuretic compounds, such as methylprednisolone, dopamine, and fenoldopam mesylate, a selective dopamine-1 receptor agonist that is responsible for quick blood pressure reduction and increasing renal blood flow. However, to date results are elusive [32,33].

The I/R following aortic cross-clamping affects not only the heart and kidney, but also the bowel, lungs, lower limbs, and spinal cord; when the damage is extensive, multi-organ failure leads to death. 

Intestinal ischemia is a rare but ominous complication after abdominal aorta surgery due to the presence of a wide network of collaterals between the inferior and upper mesenteric artery and the hypogastric arteries and branches of the deep femoral artery [34]. Ischemic complications affecting the lower limbs may be secondary to periods of hypoperfusion during aortic clamping or to embolization of fragments of the aortic wall, endoluminal thrombus, and atherosclerotic plaques during declamping. 

In routine practice, the prevention of lower limb and visceral organ ischemia caused by clamping of the aorta above the mesenteric artery is achieved by a combined technique of retrograde-aortic and selective-organ perfusion, which has been found to improve the renal and intestinal condition after I/R. The retrograde aortic perfusion provides a blood supply to the organs downstream of the distal clamping, i.e., below the aneurysm, and is obtained by cannulation of the left atrium and the femoral artery through heparin-coated tubes that are connected to a centrifugal pump, in order to ensure blood flow. The selective organ perfusion is achieved through the coeliac trunk, the superior mesenteric artery, and both renal arteries with a multi-branch catheter connected to an extra-corporal circulatory system; this procedure is described as “the octopus” perfusion technique [35].

Preclinical studies on animal models and, more recently, clinical trials, have been leading methods in order to evaluate the benefit of conditioning with brief periods of ischemia to activate cell responses against oxidative stress and inflammation as a strategy to provide protection against more prolonged ischemia.

When I/R events are predictable, such as during AAA open repair, a remote pre-conditioning before ischemia provided protection in both the heart and kidneys by reducing the absolute risk of myocardial injury by 27% and renal impairment by 23% [36].

A certain degree of protection has also been recognized with post-conditioning when the procedure is applied after the ischemic period, thus also making it applicable when I/R occurs unexpectedly [37].

As reported by Yang and colleagues [38], post-conditioning was demonstrated to be protective against ischemic changes in a canine model of myocardial ischemia and, with respect to abdominal aortic surgery on rats and mice, to counteract the inflammation and impairment of skeletal muscle and renal function. However, no clinical evidence is currently available for introducing post-conditioning into routine practice.

To date, an effective combined pharmacological treatment to protect against I/R injury is needed in major vascular surgery, such as aortic repair; in this setting, the modulation of the metabolic targets mentioned in the next chapters would offer an option to protect the heart, kidneys, and visceral organs, thereby improving outcomes and facilitating recovery. 

## 6. Preclinical Models vs. Clinical Settings—Beyond Oxidative Stress and towards Metabolic Targets

Previous preclinical studies mainly focused on oxidative damage and proposed the use of several compounds with antioxidant properties as protective strategies against I/R injury [39,40,41,42,43]. However, the available clinical trials demonstrated that merely antioxidant preventive approaches are not very effective in humans, where metabolic dysfunctions seem to prevail from ROS damage following I/R injury [44]. In mice, it was observed that I/R injury in different organs was due to an accumulation of succinate and the formation of ROS driven by succinate; in humans, the formation of ROS from succinate does not occur [45].

Species-specific differences in terms of metabolic needs, lifespan, aging processes, and cell stress response systems may account for such discrepancies.

Demetrius [46] showed that in animal models, with a body mass smaller and a lifespan shorter than in humans, the basal metabolic rate per gram of body weight, defined as the specific rate of mass, is much higher than in men. Organisms with large mass-specific metabolic rates, such as mice, are characterized by high ROS production and a weak ability to maintain homeostasis, while humans have a lower mass-specific metabolic rate and a greater ability to maintain cellular homeostasis. Moreover, in humans, multiple risk factors and pathological conditions may coexist (atherosclerosis, hypertension, renal failure, dyslipidemia, and chronic inflammatory disorders), making the management of the I/R injury harder.

Several mediators of inflammation and vascular diseases have been demonstrated to interact with metabolism, and promising results have been obtained by employing antidiabetic treatments in an experimental animal model of I/R [47]. These data support the hypothesis of counteracting I/R injury through the modulation of such metabolic pathways.

### 6.1. Proprotein Convertase Subtilisin/Kexin Type 9

Proprotein Convertase Subtilisin/Kexin type 9 (PCSK9), a pivotal regulator of low-density lipoprotein cholesterol (LDL-C), negatively affects the infarct district and cardiac function [48]. Preclinical studies demonstrated that PCSK9 inhibition gets a better infarct size and recovery after myocardial I/R injury [49]. 

PCSK9 is mainly expressed in the liver, but the kidney is also an important extra-hepatic source, and it is upregulated in several vascular diseases [50,51]; it is also highly expressed in the colon, ileum, duodenum, kidney, brain, ischemic heart [51,52,53], and in other cells, such as vascular endothelial cells (ECs), smooth muscle cells (SMCs), and macrophages [54].

At the base of many cardiovascular diseases, there is a lack of resolution of the inflammatory response of the arterial wall, which is mainly caused by an increase in LDL [55]. 

PCSK9 is not only able to induce degradation of the LDL receptor (LDLR), but also of other cell surface receptors, such as the very low-density lipoprotein receptor (VLDLR), apolipoprotein E receptor 2 (ApoER2), LRP1 [56,57], CD36 [58], CD81 [59,60], BACE1, APLP2, APP [61], and ENaC [62], thereby affecting cholesterol homeostasis and increasing inflammatory and maladaptive cell responses. Qi and colleagues observed that PCSK9 induces platelet activation by binding to CD36, supporting the possibility that PCSK9 exerts thrombogenic effects and acts, at large, as a member of the “danger associated molecular pattern” (DAMPS) superfamily [63,64]. 

The binding of PCSK9 to LDLR is mediated by the LDLR epidermal growth factor-like repeat A (EGF-A) domain [65]. Due to the presence of different targets, PCSK9 plays a distinct role in different cellular or tissue compartments during the pathogenesis of various diseases; for example, PCSK9 targeting of ApoER2 can contribute to vascular aging and other vascular diseases. PCSK9 is present in atherosclerotic plaques and affects LDLR expression in macrophages derived from SMCs [66]. The release of PCSK9 stimulates mtDNA damage that upregulates inflammatory signals, thereby causing SMC apoptosis, which is a hallmark of atherosclerosis [67].

The circulating levels of PCSK9 are associated with multiple cardiovascular and metabolic risk factors, which often coexist in patients who are prone to I/R injury [68,69,70]:

Age: Epidemiological studies demonstrated that PCSK9 increases with age and would play a crucial role in all periods of human development [51,70,71].

Diabetes mellitus (DM): PCSK9 could be positively correlated with the onset of DM, which is characterized by insulin resistance and a relative lack of insulin [72,73,74]. 

Hypertension: The relationship between circulating levels of PCSK9 and blood pressure is controversial, due to the different inclusion criteria of previous studies. However, available evidence suggests that circulating levels of PCSK9 are more likely to be positively correlated with systolic blood pressure in women than in men. Renal parenchymal hypertension is the most usual form of secondary hypertension and PCSK9 has been reported to be continuously expressed in the kidney. 

Other risk factors: Smoking could increase the risk of heart disease by stimulating inflammatory activity and oxidative stress, resulting in the increased expression of PCSK9 [50]. 

Ricci and colleagues [56] investigated the role of PCSK9 on the inflammatory state of macrophages; they demonstrated that recombinant human PCSK9 induced a pro-inflammatory phenotype, increasing the expression of inflammatory cytokines (TNF-α, IL-1β, and IL-6) and chemokines (MCP1 and CXCL2) in human THP1 that was co-cultured with HepG2 cells overexpressing PCSK9, and in mouse bone marrow-derived macrophages (BMM). 

Apaijai and colleagues [75] studied the inflammatory effects of rats that underwent cardiac I/R on the brain and reported, for the first time, the association between PCSK9 inhibition and neuronal responses, thereby preventing the loss of the dendritic spines by reducing microglial activation and the aggregation of beta-amyloid peptides.

Finally, PCSK9 levels have been shown to increase at 1–6 months after kidney transplantation, while IL-6 levels and conventional inflammatory biomarkers decrease over the same period compared to pre-transplant levels [76], indicating that PCSK9 could be a new target pathway in the first post-kidney transplant period.

### 6.2. Myostatin

Myostatin (MSTN) is a TGF-superfamily member with anti-anabolic function; it is mainly produced by skeletal muscle and, in a smaller percentage, by smooth muscle, the myocardium, and the brain in response to several stimuli, including oxidative stress, inflammation, and hypertension [77]. A known playmaker in cachexia and sarcopenia, it has been found to be upregulated in skeletal muscle cells exposed to oxidative stress conditions in the context of chronic obstructive pulmonary disease [78]. MSTN is also highly expressed 12 h after myocardial infarction in human hearts and 10 min after ischemia in the hearts and blood of mice, where it contributed to skeletal muscle atrophy by upregulating atrogin and muscle RING-finger protein. 

In a previous study, our group investigated the role of MSTN and PCSK9 in the context of renal I/R in a rat model, finding that MSTN and PCSK9 are induced early during surgical ischemia in rat kidneys and in HK-2 tubular renal cells exposed to I/R-like conditions and associated to mitochondrial impairment [48].

### 6.3. Adipokines 

Adipose tissue constitutes a central node in the inter-organ crosstalk network and mediates the regulation of multiple organs and tissues through adipokines (also called adipocytokines), which are biologically active molecules causing pleiotropic effects, including the modulation of angiogenesis, metabolism, and inflammation [79,80]. Adipokines play an important role in regulating vascular tone [81]. For instance, the adipokines adiponectin, visfatin, omentin, and the unidentified adipocyte-derived relaxing factor (ADRF) exert vasorelaxing effects on the vascular wall [82]. On the other hand, resistin and angiotensin II (AngII), which are also released by adipocytes, exert vasoconstricting effects [82]. In addition, adipose tissue-secreted leptin, ROS, apelin, TNF-α, and IL-6 share both vasoconstricting and vasorelaxing properties [83].

Upon binding to T-cadherin, adiponectin has been shown to reduce ROS production, TNF-α levels, and to protect against myocardial I/R injury [84] by activating AMPK and COX-2, respectively, with anti-apoptotic and anti-inflammatory actions [81,85].

Leptin in a healthy state controls blood pressure by regulating sympathetic activity-dependent vasoconstriction and the endothelial release of nitric oxide (NO), as well as Ang II-dependent vasoconstriction [83]. The physiological activity of leptin strongly depends on its binding to obesity receptors [81] as well as circulating leptin levels, which are positively correlated with circulating levels of PCSK9 in healthy individuals [86]. 

Resistin is a cysteine-rich, adipose-derived peptide hormone encoded by the RETN gene that is highly expressed in circulating monocytes, macrophages, and the vascular endothelium [87,88,89]. It is involved in numerous pathological processes, including atherosclerosis [90,91,92,93]. Resistin has been suggested as a marker of the severity of myocardial ischemic lesion [89,92] and has been proposed as a mediator of endothelial dysfunction [89,92,94]. Over time, elevated levels of resistin have been associated with an increased risk of coronary heart disease, especially with myocardial infarction (but not with stroke) and with the degree of heart failure, which are both responsible for increasing the rate of cardiac events, including the risk of death [95].

However, controversial results were obtained in mice. After brain I/R, the exogenous administration of resistin exerted anti-apoptotic, protective activity by decreasing the expression of Bax protein, demonstrating the ability for resistin to cross the impaired BBB after ischemia injury [96].

Some studies have indicated that circulating resistin is associated with an increased risk of acute ischemic stroke [97,98].

A close correlation between high serum resistin levels, cardiac comorbidities, and post-CA shock has been demonstrated, where the impact of post-CA shock on the serum resistin concentration was greater than that of cardiac comorbidities [99]. 

### 6.4. Peroxisome Proliferator-Activated Receptors

The Peroxisome Proliferator-Activated Receptors (PPARs) are part of the superfamily of ligand-activated nuclear receptors that bind to specific DNA regulatory elements, forming heterodimers from the enabled interaction with the retinoid X receptor (RXR) [100]. PPAR-α and β/δ are known to be significantly expressed in cardiomyocytes and are thought to play a major role in regulating fatty acid uptake and the expression of fatty acid oxidation genes [101,102]. Interestingly, both PPAR-α and β/δ have been demonstrated to be cardioprotective during I/R injury by modulating PI3K/Akt and NO [103], thereby decreasing inflammatory cytokines and upregulating pro-survival signaling such as Bcl-2 and Akt [103]. The role of PPAR-γ in the heart still appears to remain largely elusive. PPAR-γ has undoubtedly received the highest attention in the literature regarding its pronounced insulin sensitizing abilities and beneficial, although controversial, effects on the heart [104]. Moreover, PPAR-γ activation is associated with pleiotropic functions in the vasculature such as anti-inflammatory, antioxidative, anti-apoptotic, and anti-hypertensive effects [105]; PPAR-γ has been extensively reported to have various cardioprotective capabilities against I/R injury [106]. Interestingly, compared to the rest of the PPARs, PPAR-γ is expressed at the lowest abundance in cardiomyocytes [107] and has been reported to only reach approximately 30% of the expression level compared to its expression in adipocytes [108].

## 7. Pharmacological Therapies to Modulate Metabolic Targets (PCSK9, MSTN, Adipokines) and PPAR Agonists

The aforementioned molecules, which have been widely investigated over the past two decades as targets related to metabolic, cardiovascular, and inflammatory disorders, are now newly recognized playmakers in different settings of I/R injury.

With this perspective, pharmacological modulators of such metabolic targets may provide hints to define novel strategies to restrain I/R damage in the clinical context (Table 1).

Among the therapies for the treatment of hypercholesterolemia and cardiovascular diseases, the inhibition of PCSK9 has emerged [109]. As effective and safe compounds, the action of PCSK9 inhibitors relies on the engagement of the PCSK9 molecules, preventing them from binding to LDLR and provoking its degradation; as a consequence, a higher amount of oxidized LDLs is absorbed from the bloodstream, reducing their plasma level [110]. The PCSK9 inhibitors are increasingly used in patients who receive intensive lipid-lowering therapy to reduce the risk of cardiovascular events. Although some of the mechanisms by which these drugs reduce cardiovascular risk in patients are not yet known and require further investigation, the beneficial effects of their use remain unquestionable [111] (Figure 5*)*.

PCSK9 is tackled with both monoclonal antibodies and synthetic inhibitors [49,112,113]. Preclinical studies demonstrated that treatment with the synthetic PCSK9 inhibitor PEP 2-8 is effective in reducing the biochemical and physiological complications and the infarct size after cardiac I/R injury, but only when administered in the pre-ischemic phase, while it failed when it was administered during reperfusion [49]. The reason for this may reside in the fact that PCSK9 had already been released and bound to its receptors during myocardial ischemia. 

Qi and colleagues [63] found that PCSK9 inhibition was able to counteract arterial thrombosis in individuals with elevated levels of PCSK9 that were related to genetic or acquired causes. Basiak and colleagues [111] sought to systematically review current scientific data regarding the main PCSK9 inhibitors: alirocumab (human IgG1 monoclonal antibody, genetically modified in Chinese hamster ovarian cells), evolocumab (fully human monoclonal antibody), and bococizumab (humanized mouse antibody). Alirocumab and evolocumab, which are approved by the Food and Drug Administration (FDA), are mainly used in the treatment of autosomal familial hypercholesterolemia [114] and in the case of patients with statin intolerance [115]. 

Preclinical studies for the treatment of I/R injury have also been performed with inhibitors of other metabolic targets that were mentioned above; among them are leptin inhibitors. The restraint of leptin activity by an intra-arterial anti-leptin therapy may have clinical potential in reducing hemispheric brain I/R injury [116]; however, controversial results were reported, as other authors found that leptin provides neuroprotection against neuronal I/R injury and reduces I/R-induced TNF-α production in kidneys. Further investigations are needed in this regard [117,118]. 

In recent years, there has been growing evidence that PPAR agonists could be useful for the prevention of CVD. Bezafibrate is a high-affinity ligand for all three PPAR isoforms and is considered a pan-PPAR agonist [119,120]. In the in vivo study of Khazaei and colleagues, bezafibrate was found to restore angiogenesis and to prevent peripheral artery disease in ischemic diabetic rats [121].

Several in vivo studies demonstrated that PPAR-γ agonists can potentially alleviate I/R injury. In rats subjected to renal I/R, pioglitazone was shown to improve the histopathological and biological parameters by inhibiting the NF-κB pathway and inflammatory response [122]. In a model of myocardial I/R injury, the PPAR-γ agonist, rosiglitazone, reduced the infarct and ischemic sizes and improved the ventricular remodeling and recovery of cardiac function [123]. 

In 2020, Shehata and colleagues [124] compared the effects of two different agonists, PPAR-α (fenofibrate) and PPAR-γ (pioglitazone), alone or in combination, with respect to transient carotid occlusion in rats. Both of them mitigated neurobehavioral dysfunction, reduced the volume of cerebral infarction, attenuated inflammatory and apoptotic markers, and improved histopathological changes in rats affected by I/R; no additional benefits were provided by their combination. Other studies have reported that the PPAR-γ agonists alleviate lung and gastrointestinal I/R injury [125,126,127].

**Table 1 jcm-11-03638-t001:** PCSK9 inhibitors and PPAR agonists approved by FDA.

Drug [128,129]	Outcomes	Side Effects
**Evolocumab** PCSK9 inhibitor; monoclonal antibody	LDL (↓)	nasopharyngitis upper respiratory tract infection backache
**Alirocumab** PCSK9 inhibitor; monoclonal antibody	LDL (↓)	rhinopharyngitis flu-like symptoms infections of the urinary tract diarrhea bronchitis and cough muscle pain
**Pioglitazone** PPAR-γ agonist	triglycerides (↓) LDL (↓) glycemia (↓)	peripheral edema heart insufficiency bladder cancer
	HDL (↑)	
**Rosiglitazone** PPAR-γ agonist	triglycerides (↓) LDL (↓) glycemia (↓)	peripheral edema heart insufficiency
	HDL (↑)	
**Fenofibrate** PPAR-α agonist	triglycerides (↓)	rhabdomyolysis
	HDL (↑)	

Preclinical studies with anti-MSTN treatments (anti-MSTN antibodies, MSTN propeptide with inhibitory function, inhibitors of the MSTN receptor, activin-receptor IIB) have been conducted, providing encouraging results in the field of skeletal muscular dystrophy, aging and cancer-related cachexia. However, these compounds failed when used in humans. Such a discrepancy may be attributable to the following reasons: a different MSTN metabolism in human versus mice as well as between control subjects and patients’ populations; or a too broad inhibition of MSTN or of its receptor that might affect the signaling of several TGF-β superfamily members, perturbating the homeostasis cell program [130]. The generation of the selective MSTN antibody GYM329, created with the “sweeping antibody technology” [131], is still to be proven in humans and is now fueling new expectations. To date, no anti-MSTN treatment has been approved for clinical purposes by the FDA.

## 8. New Preclinical Strategies against Ischemia/Reperfusion Injury

Over the last few decades, bioengineering has developed innovative strategies for the theranostics of I/R. These futuristic tools comprise non-coding RNAs, nanoparticles (NPs), and biomaterial-based systems that have, in some cases, reached preclinical phase studies as diagnostics and/or therapeutics. A novel concept of therapy towards I/R injury is outlined by epigenetics, which considers non-coding RNAs as potential tools for targetting I/R molecular effectors [132]. Non-coding RNAs comprise short non-coding RNA, e.g., microRNAs (miRNAs), silencing RNAs (siRNAS), PIWI-interacting RNAs (piRNAs), and long non-coding RNAs (lncRNAs) [133]. The role of non-coding RNAs in I/R injury is embodied by the modulation of relevant signaling pathways that are critical in necrosis, apoptosis, inflammation, neoangiogenesis, oxidative stress, and fibrosis. Strategies based on non-coding RNAs that identify therapeutic targets such as PCSK9, MSTN, adipokines, and PPARs are not yet deeply studied, thus paving the way for new research fields. It has been found that the inhibition of PCSK9 by transfecting cardiomyocytes with siRNA induced an attenuation of LC3-II and beclin-1 expression, which are well-known markers of autophagy in response to ischemia [71]. Analogously, Huang et al. [134] demonstrated that PCSK9 knockdown by siRNA reduced myocardial I/R injury via the BNIP-3 mediated autophagic pathway and improved myocardial infarct size and cardiac function. New challenges remain to identify non-coding RNAs that, as reported for other diseases, could have a crucial role even for counteracting I/R. Several miRNAs have been considered as potential clinical biomarkers associated with ischemic diseases and were quantified in cerebrospinal fluid blood, plasma, PBMCs, and serum [135], giving them the chance to be applied, even in diagnostics. The possibility to encapsulate bioactive molecules and get to their definition as systems that present different advantages, i.e., improved pharmacokinetics, presence of targeting activity, negligible systemic toxicity, and high stability, could confine I/R damages. NPs are good candidates because for their small dimensions could penetrate into the tissue, giving a more realistic view of its architecture [136]. Specifically, for I/R, NP-based therapy and diagnosis exploits the altered vascular permeability, giving rise to an enhanced retention of NPs at the injury site. To date, different approaches have been used for the design, the production, the characterization, and the validation of NPs to prevent I/R damages. Different types of nanoconstructs have been addressed for the therapy and diagnosis of this pathological status, including lipid-based NPs (i.e., nanoliposomes and niosomes), polymeric NPs, micelles (i.e., those made of poly (lactic-*co*-glycolic acid) and chitosan), and inorganic NPs (i.e., those based on cerium and selenium). Great relevance has been attributed to polymeric-based systems for the encapsulation of PPAR-γ agonists. Tokutome et al. [137] injected poly(lactic acid/glycolic acid) NPs loaded with pioglitazone and marked with fluorescein isothiocyanate, which, as a consequence, caused the suppression of the recruitment of inflammation-related monocytes and cardioprotection from I/R injury. The same NPs in a murine model of hindlimb ischemia were responsible for an increase in blood flow recovery in the ischemic limbs [138]. As before, in order to counteract the CVD-related inflammation, Giacalone et al. [139] demonstrated the capability of poly(lactic acid) associated with poly(ethylene glycol) loaded with rosiglitazone to counteract the inflammatory process. A sine qua non for NP accumulation in the damaged site is the selective target to drive their recruitment. Different examples of targeting are reported in the literature. Peptide–polymer NPs have been prepared in a way that resulted in highly selective for matrix metalloproteinases-2, which is overexpressed in heart tissue after myocardial infarction [140]. Folic acid was considered as a target in the development of a peptide-based nanocarrier for the treatment or renal I/R injury and for the near-infrared imaging [141], whereas the soluble stromal cell-derived factor-1 was targeted by biomimetic smart NPs loaded with rapamycin to counteract cerebral I/R [142]. By contrast, feeble attempts at the employment of biomaterials to this issue are reported. Shin et al., [143] in order to reduce the innate inflammatory response during myocardial I/R injury, encapsulated rat mesenchymal stromal cells (MSCs) in alginate and put them in the anterior wall of the myocardium by using a hydrogel of poly(ethylene glycol). This combination of biomaterials ensured the good retention and viability of MSCs that exerted an anti-inflammatory effect by the conversion of AMP into adenosine. Deng et al. [144] developed a self-assembly supramolecular hydrogel that was able to release NO and curcumin at the same time. Apart from studying the inhibition of ROS-related pathways, they demonstrated that the hydrogel was responsible for reducing apoptosis and over-stimulated autophagy during I/R.

## 9. Conclusions

The considerations arisen so far indicate that the management of transient ischemic conditions still needs novel strategies to restrain the fearsome consequences of I/R injury, possibly by tackling the unbalanced metabolic flow. This goal may be achieved by protecting mitochondria when I/R events are predictable, such as during visceral surgical procedures, or in combination with percutaneous artery revascularization and anticoagulant and antiaggregant therapies when it occurs unexpectedly, as in the case of myocardial infarction, ischemic stroke, and arterial embolism. The discussed metabolic targets (PCSK9, MSTN, adipokines, and PPARs) therefore may represent new damage signals; existing studies on their modulation will provide a start-up platform to design innovative and combined strategies for personalized protection against late onset complications from I/R injury in myocardial infarction and major vascular surgery.

## Figures and Tables

**Figure 1 jcm-11-03638-f001:**
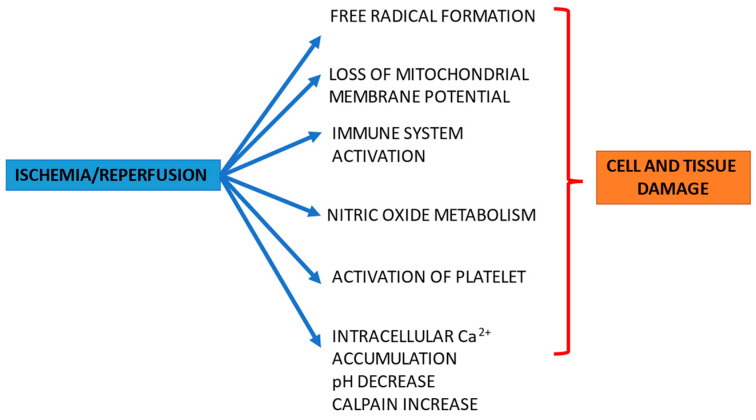
Pathological processes associated with ischemia/reperfusion in cells and tissues.

**Figure 3 jcm-11-03638-f003:**
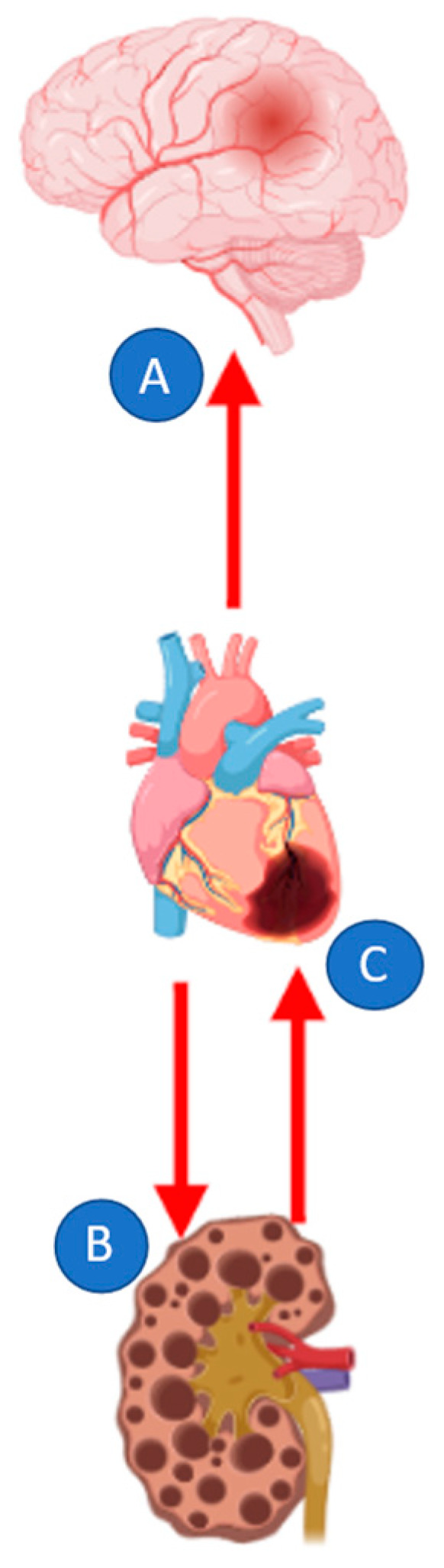
Acute myocardial infarction and heart/brain/kidney axis. (**A**): Ischemic stroke after AMI, driven by increased fibrinogen levels, thromboembolism from left ventricle, and sympathetic activation; (**B**): Kidney injury after AMI, driven by increased ROS and renal tubular damage, hypoperfusion, and systemic inflammation; (**C**): Heart disease in CKD: increased inflammation, ROS production, accumulation of vasoactive uremic toxins, and ER stress.

**Figure 4 jcm-11-03638-f004:**
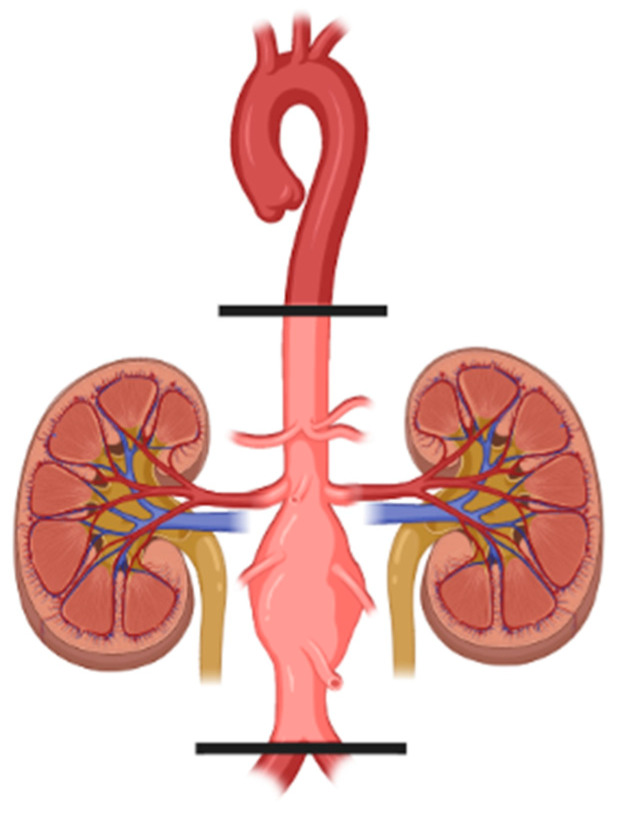
Surgically-induced ischemia during abdominal aortic aneurysm repair.

**Figure 5 jcm-11-03638-f005:**
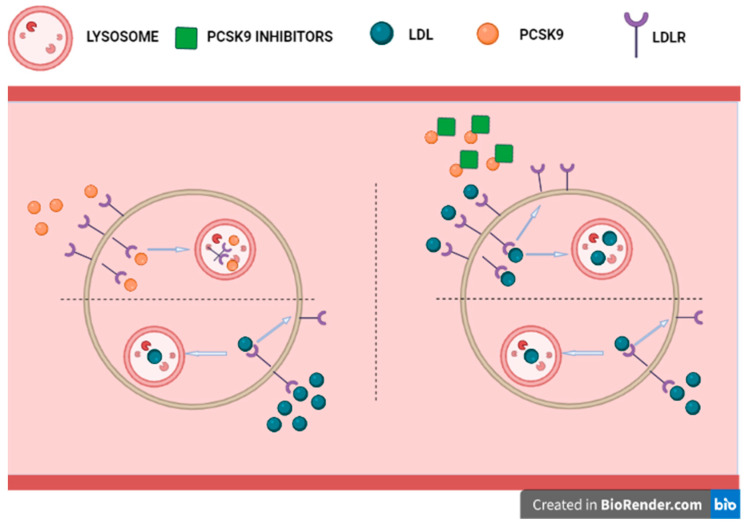
Mechanism for protection of anti-PCSK9.

## Data Availability

No new data were created or analyzed in this study. Data sharing is not applicable to this article.

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
