# Peer review of "PCSK9 and Other Metabolic Targets to Counteract Ischemia/Reperfusion Injury in Acute Myocardial Infarction and Visceral Vascular Surgery"

_jcm, 2022, doi:10.3390/jcm11133638_

Round 1

Reviewer 1 Report

Dear Authors

The paper proposed to me for review concerns the difficult clinical issue of ischemia/reperfusion in myocardial infarction and vascular surgery. The topic is topical and the work is interesting. I propose to introduce a few corrections: 1. I suggest adding a short abstract. 2. The numbering of the subsections of the articles is disordered. Please correct. 3. I propose to introduce editorial corrections; - page 2 - aligning lines 54-57 - page 5 and 6 - the comment has been moved to Figure 3 under the figure - page 13 - text transferred from page 14 - References need to be corrected in line with the Guidelines for Authors I have no substantive comments to the manuscript. Regards

Author Response

We are really grateful to the Reviewer for his/her careful revision. We did our best to fulfill the required improvements. We added a short abstract,  double-checked and corrected the highlighted editorial mistakes.  

The correction have been added to the current, revised version of our manuscript.

Best regards, on behalf the Authors, Chiara Barisione

Reviewer 2 Report

---et al .  discussed metabolic targets (PCSK9, 557 MSTN, adipokines, and PPARs) that may represent new damage signals; and their modulation  that could  provide a starting up platform to design innovative and combined strategies for personalized protection against late onset complication of I/R in- 560 jury in myocardial infarction and major vascular surgery. it is an interesting study but the following concerns should be added

1- introduction should be summarized and more consise

2-method ,details of literature search(duration, sources and keyword) should be added

3-role of Epigenetic Regulation Using MicroRNAs and Long Noncoding RNAs in IR should be highlighted

Author Response

As suggested by the Reviewer #2:

1) we made the introduction section shorter, in the attempt to render the reading smoother and more focused on our specific topic;

2) we added a "Method" section at the end of the manuscript, after "Conclusion";

3) we expanded the section 7 "New preclinical strategies against ischemia/reperfusion injury" with the appropriate consideration about the role of Epigenetic Regulation Using MicroRNAs and Long Noncoding RNAs in IR.

We really appreciated the Reviewer's suggestion to improve the quality and the novelty of our Review.

Best regards, on behalf the Authors, Chiara Barisione